# Interplay of hot electrons from localized and propagating plasmons

Chung V. Hoang[1,2,3], Koki Hayashi[1], Yasuo Ito[1], Naoki Gorai[1], Giles Allison[1], Xu Shi [4], Quan Sun [4], Zhenzhou Cheng [5], Kosei Ueno [4], Keisuke Goda[5,6] & Hiroaki Misawa [4,7]

Plasmon-induced hot-electron generation has recently received considerable interest and has been studied to develop novel applications in optoelectronics, photovoltaics and green chemistry. Such hot electrons are typically generated from either localized plasmons in metal nanoparticles or propagating plasmons in patterned metal nanostructures. Here we simultaneously generate these heterogeneous plasmon-induced hot electrons and exploit their cooperative interplay in a single metal-semiconductor device to demonstrate, as an example, wavelength-controlled polarity-switchable photoconductivity. Specifically, the dual-plasmon device produces a net photocurrent whose polarity is determined by the balance in population and directionality between the hot electrons from localized and propagating plasmons. The current responsivity and polarity-switching wavelength of the device can be varied over the entire visible spectrum by tailoring the hot-electron interplay in various ways. This phenomenon may provide flexibility to manipulate the electrical output from light-matter interaction and offer opportunities for biosensors, long-distance communications, and photoconversion applications.

[1] IMRA Japan Co., Ltd., Sapporo 004-0015, Japan. [2] Institute of Materials Science, Vietnam Academy of Science and Technology, Hanoi, Vietnam. [3] Faculty of Engineering Physics and Nanotechnology, VNU University of Engineering and Technology, Hanoi, Vietnam. [4] Research Institute for Electronic Science, Hokkaido University, Sapporo 001-0021, Japan. [5] Department of Chemistry, University of Tokyo, Tokyo 113-0033, Japan. [6] Department of Electrical Engineering, University of California, Los Angeles, CA 90095, USA. [7] Department of Applied Chemistry, National Chiao Tung University, Hsinchu 30010, Taiwan. Correspondence and requests for materials should be addressed to C.V.H. (email: chunghv@ims.vast.ac.vn) or to H.M. (email: misawa@es.hokudai.ac.jp)

In recent years, plasmons or coherent oscillations of free electrons in metals have attracted a great deal of attention from researchers in a diverse range of areas such as molecular spectroscopy[1–3], biomedical sensing[4–6], and solar energy harvesting[7–17]. To realize applications in these areas, plasmons have been studied and used for electromagnetic field enhancement[18–26], spectral-response engineering[27–30], and high-efficiency photothermal conversion[18, 31, 32]. Plasmons can be launched via the electromagnetic coupling between incident light and free electrons in either metal nanoparticles (NPs) or infinite metal films; these phenomena are known as localized surface plasmon resonance (LSPR)[8] and surface plasmon polaritons (SPPs)[33], respectively. After being launched, plasmons decay via a radiative or non-radiative process. In the radiative process, light is radiated by the metals acting as optical antennas[24, 25, 30] whereas, in the non-radiative process, hot electrons are generated from the metals and can be harvested by Schottky junctions[7–15, 34–37].

Among various classes of plasmonic systems, film-coupled plasmonic nanostructures exhibit intriguing effects that are setting a basis for groundbreaking innovations in many areas[18, 26, 36, 38, 39]. In such systems, the metal NPs which support LSPR effectively couple with the metal film, allowing for the strong confinement of incident light within a tiny volume with dimensions much smaller than the wavelength of light[39]. In the field of optics, this coupling offers unprecedented opportunities for efficiently trapping and manipulating light at the nanoscale. This effect has immediate relevance for designing perfect absorbers[18], tailoring optical spectra[26, 36, 38], and enhancing molecular vibrational spectroscopy[39]. However, in the field of optoelectronics where the harvest and control of the optically excited charge carriers are important, previous efforts have been predominantly focused on manipulating light absorption and elucidating the dynamics of hot-electron generation via plasmon damping of the LSPR or SPPs as separate processes only[8, 23, 35]. Consequently, the process of hot-electron generation in film-coupled plasmonic nanostructures still remains to be understood. A better understanding of this process is expected to foster the development of new technologies for generating and controlling plasmon-induced hot-electrons and to facilitate the development of novel plasmonic devices.

In this article, we study these two types of plasmon-induced hot-electron generation in a single metal-semiconductor device and demonstrate the cooperative interplay between hot electrons generated from both plasmonic NPs (localized plasmons: LSPR) and a thin metal film (propagating plasmons: SPPs). Specifically, our dual-plasmon device consists of AuNPs embedded in a TiO$_2$ layer on top of an infinite Au film. The device produces a net photocurrent whose polarity is determined by the balance in population and directionality between the hot electrons from the plasmonic AuNPs and the infinite Au film, depending on whether the excitation wavelength is on the LSPR of the AuNPs (Supplementary Movies 1 and 2, Supplementary Notes 1 and 2). While the LSPR originates from the AuNPs, the SPPs are launched on the Au film via scattered light from the AuNPs. By virtue of this design, the resulting LSPR and SPPs effects are of a cooperative nature rather than merely being the sum of the two individual plasmonic effects. Numerous applications can result from this effect and its tunability, one of them being, for example, wavelength-controlled polarity-switchable photoconductivity as experimentally demonstrated below. Given the simplicity of the device, it may be useful as a replacement for current optical instruments such as wavemeters as well as other devices that hold complex functionalities, but suffer from a bulky and impractical setup. The presented dual-plasmon device may allow for the miniaturization and simplified control of such setups and enhance the applicability and accessibility of optoelectronic methods to research areas such as bio- and photo-chemistry.

## Results

**Principles of the dual-plasmon device.** As schematically shown in Fig. 1a, the dual-plasmon device is composed of a 100 μm thick water layer, a 25 nm thick TiO$_2$ active layer in which AuNPs (about 4 nm in diameter) are embedded at a distance of 5 nm below the water layer, and an ~ 200 nm optically thick, 3.14 mm$^2$ large Au film below the TiO$_2$ layer (see Methods for details). The 5 nm thick TiO$_2$ cover on top of the AuNPs/TiO$_2$ layer is used to eliminate possible charge transfer from the AuNPs to the 0.1 M KClO$_4$ electrolyte solution. Because the AuNPs are not in direct contact with the water layer, they neither act as catalyst centers for the recombination of hydrogen atoms nor contribute to the redox reaction in the device[40]. Furthermore, the AuNPs are embedded in the TiO$_2$ layer to substantially increase the efficiency of the hot-electron transfer from the AuNPs to the TiO$_2$ layer[41]. The Au film is used to launch SPPs at the interface between the TiO$_2$ layer and the Au film. A transmission electron microscope (TEM) image of the fabricated dual-plasmon device is shown in Fig. 1b.

A band diagram of the dual-plasmon device is shown in Fig. 1c. The device is used as a working electrode in a photoelectrochemical measurement configuration with water as an aqueous electrolyte (Supplementary Fig. 3, Supplementary Note 3). Illumination with visible light excites electrons via interband transitions ($\lambda < 500$ nm) or intraband transitions (500 nm $< \lambda <$ 700 nm) in the Au film. Excited electrons can directly surpass the Au/TiO$_2$ barrier (about 1 eV)[7, 11] and contribute to the net

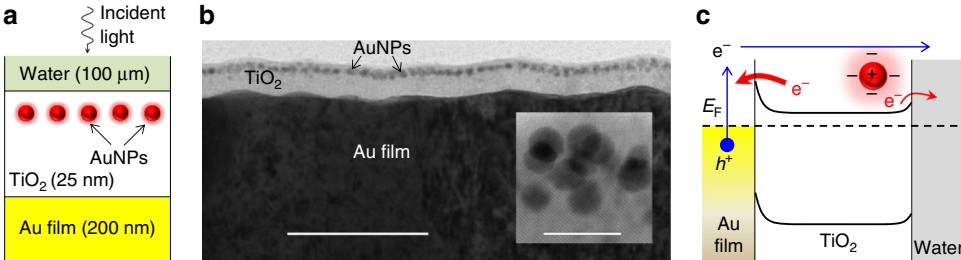

**Fig. 1** Schematic and working principle of the dual-plasmon device. **a** Schematic of the device. **b** Cross-sectional transmission electron microscope (TEM) image of the device, *scale bar* 100 nm. The inset shows the Au nanoparticles (AuNPs) at a higher magnification, *scale bar* 10 nm. **c** Working principle of the device under the optical excitation of the localized surface plasmon resonance (LSPR) of the AuNPs. The *blue arrows* indicate the transfer of hot electrons generated from a random surface plasmon polariton (SPP) in the Au film and partially from the interband or intraband transitions of the Au film to the water layer. The *red arrows* indicate the transfer of hot electrons generated from the AuNPs to the Au film and water layer. The population of hot electrons that are transferred from the AuNPs to the Au film is larger than that to the water layer, owing to the requirement of the redox reaction, indicated by different thicknesses of the red arrows

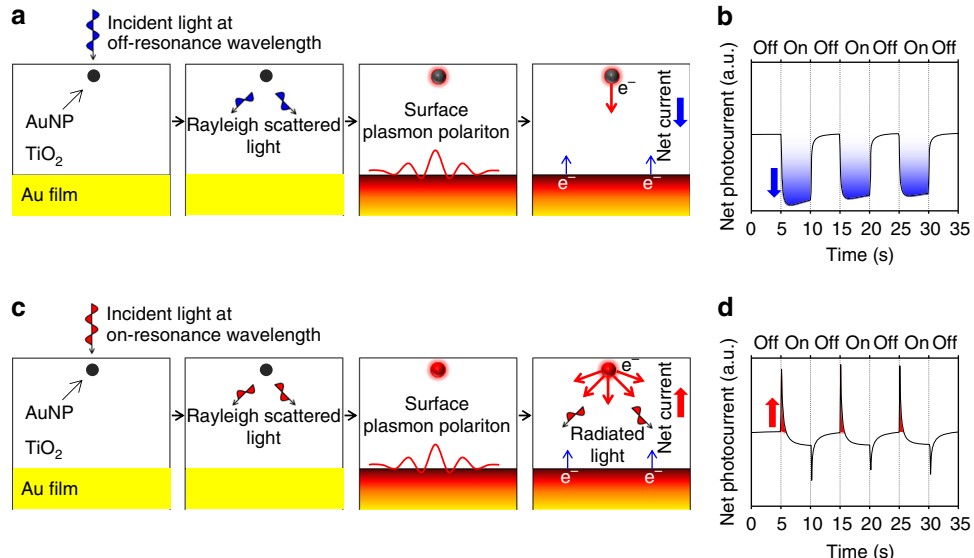

**Fig. 2** Generation and interplay of heterogeneous hot electrons in the dual-plasmon device under two characteristic conditions and the corresponding experimental demonstrations. **a** Au nanoparticle (AuNP) embedded in a $TiO_2$ layer on top of an Au film, which is illuminated by light whose wavelength is off-resonance with the localized surface plasmon resonance (LSPR) of the AuNP. The incident light is Rayleigh-scattered by the AuNP. The scattered light excites the Au film to generate a random surface plasmon polariton (SPP) at its interface with the $TiO_2$ layer, which subsequently emits hot electrons via non-radiative plasmon decay. Simultaneously, the AuNP emits a small number of hot electrons by non-radiative plasmon decay. The net photocurrent is negative. **b** Experimental verification for the process in **a** by our typical measurements when the illumination is turned on and off at 600 nm (off-resonance with the LSPR). **c** Embedded AuNP in a $TiO_2$ layer on top of an Au film illuminated by light whose wavelength is on-resonance with the LSPR of the AuNP. As in the first case, a random SPP is generated by the scattered light and decays by emitting hot electrons. Simultaneously, the AuNP emits a large number of hot electrons via non-radiative plasmon decay and omnidirectional light via radiative plasmon decay, which also induces the generation of another SPP. The net photocurrent is positive. Here, the direct excitation of hot electrons by the interband or intraband transitions in the Au film is not presented as it is common to two cases. **d** Experimental verification for the process in **c** by our typical measurements when the illumination is turned on and off at 700 nm (on-resonance with the LSPR). See Supplementary Movies 1 and 2 for the animation of the hot-electron interplay and the experimental demonstration, respectively

photocurrent via a reduction reaction at the $TiO_2$/water interface[42–44]. Because the AuNPs are spherically shaped and sufficiently small, Rayleigh scattering comes into effect under the illumination. This elastic scattering offers larger scattering angles than Mie scattering and ensures momentum conservation at the Au/$TiO_2$ interface to launch SPPs. The Au film generates hot electrons toward the $TiO_2$ layer via SPP decay, which contributes to the negative net photocurrent (cathodic photocurrent). During this process, the electrons can be partially intercepted by the AuNPs embedded in the $TiO_2$ which reduces the quantum efficiency. On the other hand, the plasmonically excited AuNPs operate as optical antennas, which emit omnidirectional light toward the Au film via radiative plasmon damping. The non-radiative plasmon damping of the AuNPs generates hot electrons toward the $TiO_2$ layer via a Schottky barrier between the AuNPs and $TiO_2$ layer. The hot electrons generated from the AuNPs can be transferred to the Au film and water layer, although the latter process occurs with a lower probability than the former process as it requires a reduction reaction at the $TiO_2$/water interface[42–44].

**Generation and interplay of heterogeneous hot electrons**. Two representative cases of the functionality of the dual-plasmon device are schematically depicted and experimentally verified, as shown in Fig. 2. For simplicity, the direct excitation of hot electrons by the interband or intraband transitions in the Au film is not presented in the figures as it is common in both cases. Figure 2a shows the functionality of the device illuminated by light whose wavelength is not resonant with the LSPR of the AuNP. The incident light is Rayleigh-scattered by the AuNP. The scattered light excites the Au film to generate a random SPP at its interface with $TiO_2$, which subsequently emits hot electrons via non-radiative plasmon decay. Simultaneously, the AuNP emits a small number of hot electrons via non-radiative plasmon decay. The net photocurrent is negative. Our experimental demonstration under illumination at an off-resonance wavelength of 600 nm verifies this process (Fig. 2b, Supplementary Movie 2 and Supplementary Note 2).

Figure 2c shows the functionality of the device illuminated by light whose wavelength is on the resonance with the LSPR of the AuNP. As in the first case, a random SPP is generated and decays by emitting hot electrons. Simultaneously, the AuNP emits a large number of hot electrons via non-radiative plasmon decay and omnidirectional light via radiative plasmon decay, which also induces the generation of another random SPP. The net photocurrent is positive. Finally, a state of equilibrium is reached between the generated hot electrons and the diffused hot electrons at the AuNPs such that no more charges are produced at the AuNPs. Our experimental demonstration under illumination at an on-resonance wavelength of 700 nm verifies this process (Fig. 2d, Supplementary Movie 2 and Supplementary Note 2).

**Wavelength-controlled polarity-switchable photoconductivity**. Wavelength-controlled polarity-switchable photoconductivity is one unique application of the hot-electron interplay in the dual-plasmon device. Here, the LSPR wavelength of the AuNPs is designed to be 700 nm. As shown in Fig. 3a, the measured photocurrent in the device experiences a polarity switch from negative to positive at about 650 nm, as the center wavelength of the incident light is swept from 400 to 900 nm with a bandwidth of 10 nm. While the negative photocurrent decays slowly, the decay

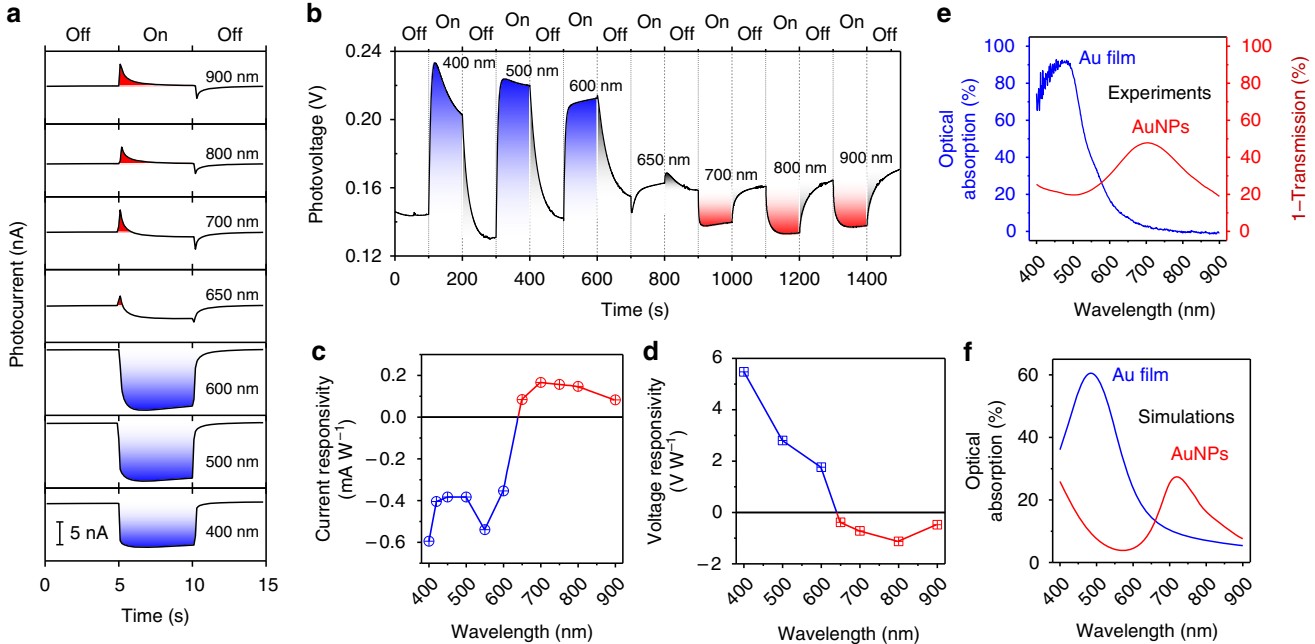

**Fig. 3** Wavelength-controlled polarity-switchable photoconductivity of the dual-plasmon device. **a** Photocurrent in the dual-plasmon device when the illumination is turned on and off and swept in the wavelength range of 400–900 nm. The polarity-switching wavelength is about 650 nm. See Supplementary Movie 2 for the experimental demonstration of the wavelength-controlled polarity-switchable photoconductance. **b** Evolution of the photovoltage when the illumination is turned on and off and continuously swept in the wavelength range of 400–900 nm. **c** Current responsivity of the device. **d** Voltage responsivity of the device. **e** Measured optical absorption of the device in the absence of the Au film (*blue*) and AuNPs (*red*). **f** Finite-difference time-domain (FDTD) simulations of the optical absorption of the device in the absence of the Au film (*blue*) and Au nanoparticles (AuNPs) (*red*)

of the positive photocurrent is burst-like, indicating that the device works like a capacitor. Here, once the hot electrons are spilled out by the internal photoemission owing to the damping of the LSPR, a local electric field is formed in the direction from the AuNPs toward the $TiO_2$ and prevents the diffusion of hot electrons in this direction: forming a local capacitor at this interface and causing the burst-like feature of the positive photocurrent. In the case of Fig. 3a, the positive photocurrent from the AuNPs and the negative photocurrent from the Au film come into play. While the positive photocurrent has a burst-like feature, the negative photocurrent is more stable under visible illumination. This interplay effect induces the strong burst-like feature of the photocurrent, as presented in Fig. 3a. Meanwhile, the hot electrons generated by the interband or intraband transitions and random SPP damping in the infinite Au film participate in the reduction reaction at the $TiO_2$/water interface.

Figure 3b shows the measured photovoltage across the dual-plasmon device, when the illumination wavelength is swept likewise, indicating a polarity switch at about 650 nm which agrees with that of the measured photocurrent. At wavelengths shorter than 650 nm, the photovoltage increases in the positive direction which corresponds to the negative net photocurrent. At these wavelengths, the increasing photovoltage indicates the transfer of the generated hot electrons from the Au film to the water, whereas the decreasing photovoltage in the absence of illumination indicates the migration of discharging or cold electrons from the water to the Au film. The response and relaxation times of the measured photovoltage are minimized at the polarity-switching wavelength on the second time scale (see Supplementary Fig. 4 and Supplementary Note 4 for details). These values are much longer than that of the plasmon lifetime which is known to be on the order of femtoseconds[45, 46]. In addition, the presence of the Au film significantly increases the relaxation time of the dual-plasmon device compared with the device without the Au film. The second time scale measured in

our device originates from various factors, including the Schottky contacts at the interfaces of the $TiO_2$/AuNPs and $TiO_2$/Au film, the electron trapping states due to defects in the $TiO_2$ layer, and the redox reaction at the $TiO_2$/water interface. It is worthwhile to mention that the polarity-switchable function of the net photocurrent is observed owing to the presence of water, which exhibits the dual property. The water layer enables this function by simultaneously supporting the reduction and oxidation reactions at the $TiO_2$/water interface[42–44]. Exchanging the water layer with a p-type semiconductor layer causes the formation of a p-n junction at this interface, since $TiO_2$ is an n-type semiconductor in nature. Consequently, the interplay effect is absent because AuNP-induced hot electrons only move in one direction, governed by the p-n junction[47].

The current and voltage responsivities of the device are shown in Fig. 3c, d, respectively. These responsivities and polarity-switching wavelength can be explained by the optical absorptivity of the device in the visible range. Figure 3e shows the optical absorption of the device in the absence of the Au film or AuNPs, which is consistent with our finite-difference time-domain (FDTD) simulation results shown in Fig. 3f (also see Supplementary Figs. 5 and 6 and Supplementary Note 5 for details). Here, at wavelengths shorter than about 650 nm[48], the intrinsic interband or intraband transitions of the Au film contribute more to the overall performance of the device than the LSPR of the AuNPs, whereas the contribution is opposite at wavelengths longer than 650 nm, which leads to the polarity switch at about 650 nm. The combination of the two absorption spectra, hence, accounts for the current and voltage responsivities.

**Photocurrent in the device without the AuNPs**. The hot-electron interplay effect is absent without the contribution of hot electrons generated by the LSPR of the AuNPs. We investigate this phenomenon by examining the functionality of three devices namely, device 1: fluorine-doped tin oxide (FTO)/Au

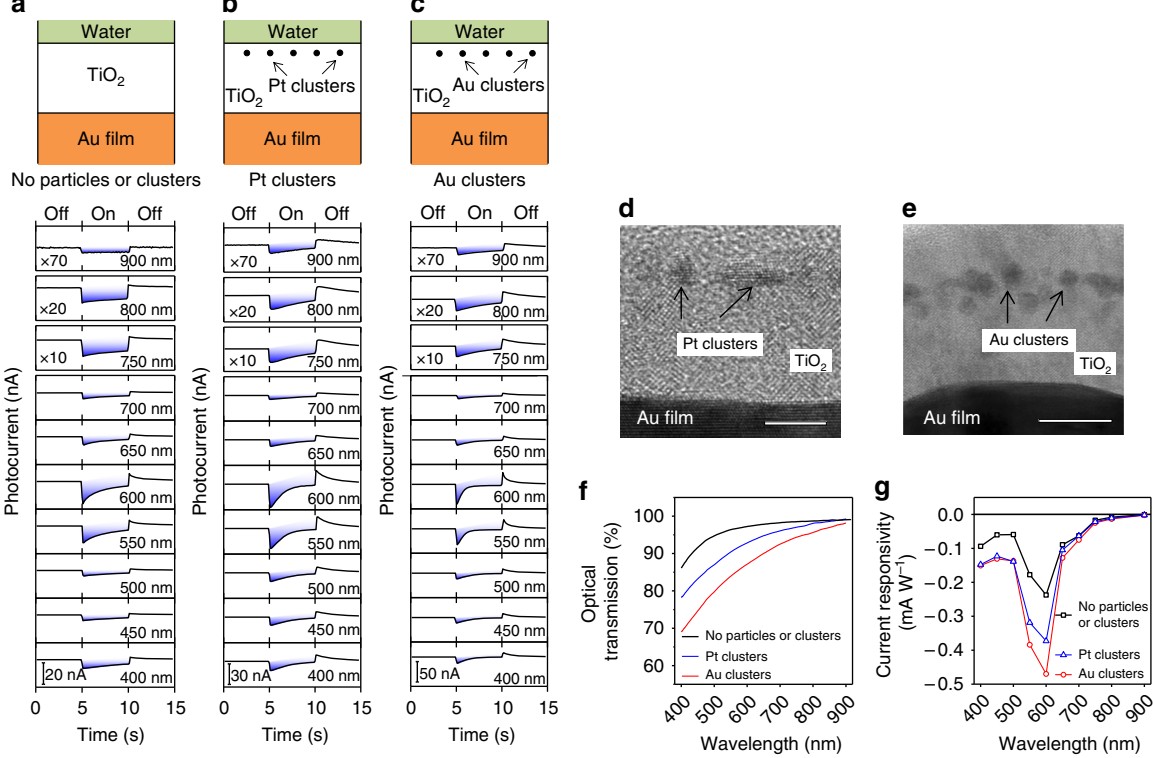

**Fig. 4** Absence of the hot-electron interplay effect without Au nanoparticles (AuNPs). **a** Schematic of the device in the absence of AuNPs and results of the photocurrent measurements. **b** Schematic of the device in which Pt clusters are embedded in place of AuNPs and results of the photocurrent measurements. **c** Schematic of the device in which Au clusters are embedded in place of AuNPs and results of the photocurrent measurements. **d**, **e** Transmission electron microscope (TEM) measurements of the Pt clusters and Au clusters. *Scale bars*: 5 and 10 nm, respectively. **f** Optical transmission through the devices in the absence of the Au film. **g** Photocurrent responsivities of the devices presented in **a**, **b**, and **c**

film/TiO$_2$ in the absence of the AuNPs (Fig. 4a); device 2: FTO/Au film/TiO$_2$/Pt clusters/TiO$_2$ (Fig. 4b); device 3: FTO/Au film/TiO$_2$/Au clusters/TiO$_2$ (Fig. 4c). The Pt and Au clusters are formed by depositing a 0.3 nm thick Pt or Au film into the TiO$_2$ layer. As shown in Fig. 4d, e, the Pt and Au clusters are formed in the TiO$_2$ layer with an average diameter of less than 3 nm and ~ 2.5 nm, respectively (the Pt clusters are less uniform in size and shape than the Au clusters because of the high melting temperature of Pt). The photocurrent measurements of the devices along with their schematics are shown in Fig. 4a–c.

The photocurrents are measured when the illumination is turned on and off at wavelengths swept from 400 to 900 nm. Over the entire illumination spectrum, only the negative net photocurrents are obtained under illumination. The negative net photocurrent in the first device (schematic illustration presented in Fig. 4a) is mainly attributed to the direct excitation of electrons via the interband or intraband transitions of the Au film. An additional influence may arise from scattered light by grain boundaries formed by small TiO$_2$ nanocrystals (Supplementary Fig. 6). It is worthwhile to note that the lineshape of the photocurrents presented in Fig. 4a shows a wavelength-dependent behavior. However, this feature also depends on the defect density of the TiO$_2$ layer. In our experiments, it is known that the TiO$_2$ layer prepared by atomic layer deposition (ALD) has a much smaller defect density than the sputtering method, and therefore the feature disappears for devices prepared by ALD (data not shown).

Similar to the net photocurrent shown in Fig. 4a, the net photocurrents shown in Fig. 4b, c are also negative, when the illumination is on at wavelengths over the same spectral range. These are attributed to the absence of an LSPR in the Pt and Au clusters. The optical transmission through these devices shown in

Fig. 4f supports this observation and indicates that these clusters do not display any similar resonant features to the LSPR of AuNPs (a typical spectrum shown in Fig. 3e)[49, 50]. Instead, they scatter incident light via Rayleigh scattering, which can fulfill momentum conservation for launching random SPPs at the TiO$_2$/Au film interface and hence produce hot electrons, resulting in a stronger negative net-photocurrent signal.

Finally, Fig. 4g compares the current responsivity of these devices and indicates that the wavelength-controlled polarity-switching photoconductivity is absent in these devices. The device with the Au clusters exhibits a two-fold larger photocurrent compared to the device without particles or clusters, meaning that the hot electrons induced by the interband or intraband transitions account for about 50% of the total negative photocurrent. In Fig. 4g, similar dips located at the wavelength of around 600 nm are observed in the photocurrent measurements for the three cases. These dips are not related to the plasmon resonance as the TiO$_2$ film with and without the clusters does not exhibit any resonance, as presented in Fig. 4f. However, the defect-induced electronic levels in the TiO$_2$ film can generate a photocurrent which partially compensates for the photocurrent from the Au film. Therefore, the dips are present in the photocurrent response. The stronger current responsivities of the second device (with Pt clusters, Fig. 4b) and third device (with Au clusters, Fig. 4c) reflect that the randomly generated SPPs play a significant role in the generation of hot electrons for the enhanced negative net photocurrent.

**Photocurrent in the device without the Au film.** The hot-electron interplay effect is also absent without the contribution of the Au film. We investigate this phenomenon by examining the functionality of the device without the Au film. As

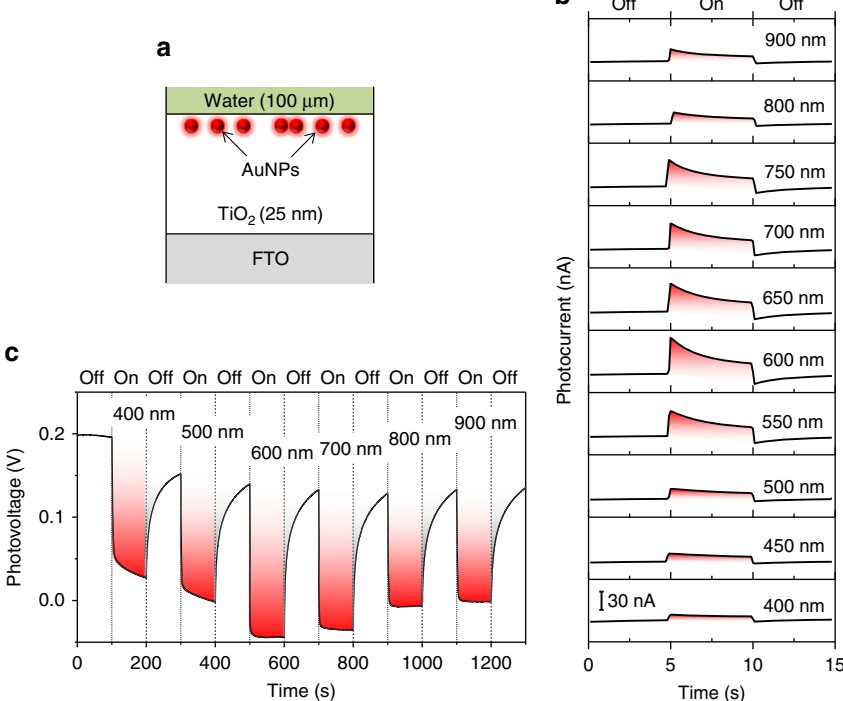

**Fig. 5** Absence of the hot-electron interplay effect without the Au film. **a** Schematic of the device in which the 200 nm thick Au film is absent. A 500 nm thick fluorine-doped tin oxide (FTO) layer on glass is used as the substrate for the fabrication of the TiO$_2$ layer. **b** Results of the photocurrent measurements. **c** Results of the photovoltage measurements

schematically shown in Fig. 5a, the device has AuNPs, but does not have the Au film below the TiO$_2$ layer. As shown in Fig. 5b, the photocurrent measurements of the device at illumination wavelengths of 400–900 nm show that the net photocurrents are always positive when the illumination is on. The corresponding negative photovoltage is shown in Fig. 5c. This type of single-polarity photoconductance without a switching effect is common and well documented in the literature[11–15]. The non-radiative plasmon damping of the LSPR of the AuNPs results in the generation of hot electrons in the conduction band and hot holes in the valence band of the AuNPs. The hot electrons surpass the Schottky barrier at the AuNP/TiO$_2$ interface and subsequently migrate to the electrode whereas the hot holes migrate to the valence band of the TiO$_2$ layer, diffuse to the TiO$_2$/water interface, and subsequently participate in the oxidation reaction of water.

**Tailoring the switching wavelength.** The current responsivity and polarity-switching wavelength of the dual-plasmon device can be varied by tailoring the hot-electron interplay over the entire visible spectrum in various ways. First, the LSPR of the AuNPs can be varied by changing their properties, namely their size, shape, dielectric properties, and surrounding medium[30]. As shown in Fig. 6a, the current responsivity and polarity-switching wavelength of the device are tunable by increasing the size of the AuNPs and consequently shifting their LSPR (Supplementary Figs. 7–9, and Supplementary Note 6). These attributes can also be changed by varying the gap between the AuNPs and the Au film. Reducing the gap while keeping the thickness of the TiO$_2$ layer intact makes the generation of hot electrons from the Au film dominant over that from the AuNPs (Fig. 6b). Moreover, the current responsivity and polarity-switching wavelength can be varied by changing the potential applied to the electrodes (Fig. 6c) or varying the pH of the electrolyte solution (Supplementary Fig. 10). Finally, the polarity-switching wavelength can be shifted

from about 580 nm to about 490 nm by reducing the Schottky barrier at the Au film/TiO$_2$ interface via an additional thin Ti layer (about 3 nm) between the Au film and the TiO$_2$ layer and making it an Ohmic-like contact (Fig. 6d, Supplementary Figs. 11 and 12 and Supplementary Note 7).

## Discussion

In summary, this work demonstrates the co-existence and cooperation of hot electrons generated by the damping of LSPR and SPPs in a metal-semiconductor device. The simultaneous generation of hot electrons by these two types of plasmons causes a cooperative interplay effect and offers the possibility of regulating the net photocurrent. We demonstrate this phenomenon with a wavelength-controlled polarity-switchable dual-plasmon device whose polarity-switching wavelength can be controlled by either changing the design of the device (passive control) or by triggering the applied bias potential (active control). It is envisioned that this work may provide the flexibility to manipulate the electrical output from the light-matter interaction and offer opportunities for bio-sensors, long-distance communications, and photoconversion applications.

## Methods

**Device fabrication.** Prior to the fabrication of the dual-plasmon device, an FTO/glass substrate ($10 \times 10 \times 2.2$ mm, Furuuchi Co., Japan) was cleaned by acetone, ethanol, and water in an ultrasonic bath for 5 min each. The dried substrate was transferred to a sputtering instrument (SH250, ULVAC, Inc. Japan) and evacuated to pressure lower than 8 μtorr. A 200-nm thick Au layer was deposited onto the conductive side of the FTO/glass substrate. Sputtering was performed in an Ar environment with a flow rate of 12 s.c.c.m. and a sputtering power of 100 W. Subsequently, a 20-nm thick TiO$_2$ layer was sputtered in an Ar environment with a flow rate of 12 s.c.c.m and a sputtering power of 100 W. An ultra-thin Au layer was deposited under similar conditions to the ones for deposition of the 200-nm thick Au layer, applying a 1.5 cm diameter hole mask to reduce the sputtering rate. For embedding the AuNPs in TiO$_2$, a 5 nm thick TiO$_2$ top layer was deposited under the same conditions as for the deposition of the 20-nm thick TiO$_2$ layer. The device was annealed in a quartz tube of an infrared furnace in a dried air

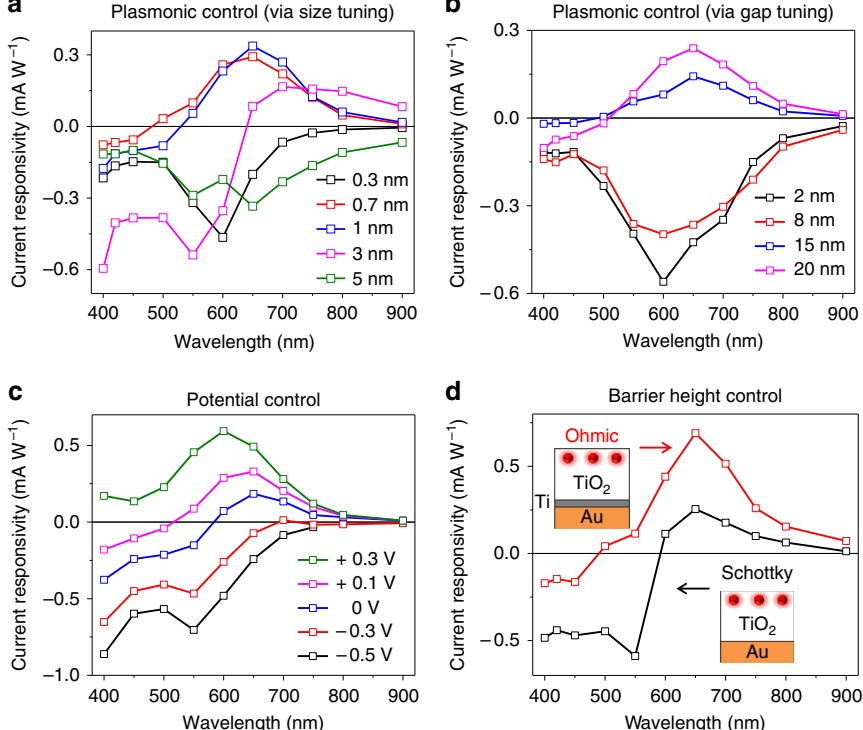

**Fig. 6** Tailoring the current responsivity and polarity-switching wavelength of the dual-plasmon device. **a** Current responsivity of the device with AuNPs of various diameters (passive plasmonic control). The values indicate the thickness of the deposited Au film before annealing it to form the AuNPs. **b** Current responsivity of the device with various gaps between the AuNPs and the Au film, whereas the thickness of the $TiO_2$ layer is kept constant (passive plasmonic control). **c** Current responsivities of the device at various applied voltage values (active potential control). **d** Effect of the Ohmic contact at the Au film/$TiO_2$ interface (passive barrier height control)

environment with a flow rate of 200 s.c.c.m. to produce the AuNPs in the $TiO_2$ layer. The temperature of the device was raised from room temperature to 450 °C within 30 min. The device was kept at the temperature for 1 h and cooled down to room temperature for 1.5 h. See Supplementary Fig. 13 and Supplementary Note 8 for the fabrication of the device used in the demonstration.

**Device characterization**. The fabricated dual-plasmon device was characterized by using an X-ray diffraction analyzer and high-resolution TEM (JEM-ARM200F, JEOL) (see Supplementary Figs. 1 and 2 for results). The functionality of the dual-plasmon device was investigated by the conventional photoelectrochemical (PEC) measurement technique (electrochemical analyzer, model 802D, CH Instruments, Inc.) in a three-electrode configuration: Pt as a counter electrode, Ag/AgCl as a reference electrode, and the dual-plasmon device as a working electrode (see Supplementary Fig. 3 for detail). In this configuration, a custom-made PEC cell was constructed in a single Pyrex glass reactor compartment (about 80% of optical transmission at $\lambda = 350$ nm and $> 90\%$ of optical transmission from 400 to 900 nm) with a salt bridge in the middle. A 0.1 M $KClO_4$ (from Kanto Chemical Co. Ltd.,) with a pH of about 7 was used as an electrolyte solution. The optical illumination was performed with a Xe lamp and various bandpass filters (with a bandwidth of 10 nm) from the visible to near-infrared band. The illumination area on the working electrode was fixed to $S = 3.14$ mm$^2$. In this configuration, the applied voltage is between the working and reference electrodes.

**Data availability statement**. The authors declare that (the/all other) data supporting the findings of this study are available within the paper (and its Supplementary Information files).

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

## Acknowledgements

We thank N. Hasegawa, H. Suzuki, M. Kobayashi, and M. Ando at IMRA Japan Co., Ltd., for their insightful discussion. A part of this work was conducted at Hokkaido University under the support of "Nanotechnology Platform" Program of the Ministry of Education, Culture, Sports, Science and Technology (MEXT), Japan.

## Author contributions

C.V.H. designed the study. C.V.H., K.H., and Y.I. performed the experiments. X.S., K.U., and H.M. provided technical support. N.G. designed the electronics to demonstrate the dual-plasmon device. Q.S., G.A., and K.U. performed the FDTD simulations. C.V.H., K.U., H.M., Z.C., and K.G. had technical discussions to theoretically understand the underlying mechanism of the interplay between the hot electrons and wrote the manuscript. All authors participated in the discussion of the manuscript. N.G., K.G., and H.M. supervised the work.

## Additional information

**Competing interests:** The authors declare no competing financial interests.

