## [Peer Review File · Nature Communications]

Reviewers' comments:

Reviewer #1 (Remarks to the Author):

Overall, this is a very nice work investigating the interplay between hot electrons generated from LSPRs and SPPs. The authors leveraged their understanding on this interplay to demonstrate a wavelength-controlled polarity-switchable photoconductivity device. To my best knowledge, this is the very first report of a device of this type. The experiments are well-designed and the data are overall well-presented and –interpreted. Given the interesting fundamental science unraveled in this work as well as some potential applications of the polarity-switchable device, I see a strong potential of this manuscript to be published as an article in Nat. Comm. Yet there are many detailed revisions that the authors might want to perform before the decision can be made.

1. The authors mentioned many times (e.g. line97-98) that the Au film could be excited by the inter-inttaband transitions. This statement is somewhat ambiguous. I remember that the interband onset of bulk Au is ~ 2.4 eV, below which the excitation is dominated by intraband transitions. The authors should present their data more rigorously according to this fact.

2. Line 98, the authors claimed that “excited electrons can directly surpass the Au – TiO₂ barrier (1.7 eV) [21] and contribute to the net photocurrent via the reduction reaction at the TiO₂ – water interface”, which to me is not rigorous. While some of the electrons can indeed travel ballistically through the TiO₂ layer, the authors have to acknowledge that the electrons can also be intercepted by the Au NPs embedded in the TiO₂. Due to the band structure, Au NPs are actually excellent electron sinks. This raises a related question. Have the authors attempted to quantify the quantum efficiency of the device? Due to the interception effect I mentioned, I suspect that the QE should be extremely low.

3. What is the purpose of Fig. 2a? Fig. 2b&c seem to make a nice comparison already. To me, Fig. 2a is redundant, especially because Fig. 4 will present the same information. Another question for Fig. 2, are the photocurrent figures real data or just schemes? I haven't seen any clarifications on this. If they are real data, why does the photocurrent decay so fast when the light is on and switch the sign when the light is off in Fig. 2a, but not in Fig. 2b?

4. Line 139-142, the authors state that “Here, once the hot electrons are spilled out by the internal photoemission owing to the damping of the LSPR, a local electric field is formed in the direction from the Au NPs toward the TiO₂ and prevents the diffusion of hot electrons in this direction, forming a local capacitor at this interface and causing the burst-like feature of the positive photocurrent”. While this is reasonable explanation for the data in Fig. 3a, why don't we the same photocurrent burst in Fig. 5. The photocurrent in Fig. 5 is solely generated by the hot electron transfer from LSPRs, which should also give a current burst.

5. What is the reason for the current resonance peak at ~ 600 nm in Fig. 4g? Here, the photocurrent either results from direct optical excitation of Au film (Fig. 4a) or scattered light excitation (Fig. 4b&c). I don't see any resonance mechanisms in these processes.

6. Fig. 6a, why don't we see the polarity switching on the curve of 5 nm Au NPs?

7. The Method section is repeated in the SI. The authors should rearrange it.

Reviewer #2 (Remarks to the Author):

The authors realized the simultaneous generation of localized and propagating plasmon-induced hot electrons and investigated their cooperative interplay in a metal-semiconductor device. And they also demonstrated such a device as a wavelength-controlled polarity-switchable photoconductivity. According to theoretical analyses and experimental verifications, they elucidated that the polarity of the dual-plasmon device is determined by the balance in population and directionality between the hot electrons generated from LSPR and SPPs. Also, they demonstrated that the functionality and

performance can be tailored by changing the design of the device or the applied bias potential.

It is a good and novel work, both experiments and theoretical analyses are sufficient. The presented results are interesting. Therefore, I will recommend this paper to be published in Nature Communications with possibly minor revisions based on the comment below:

(1) In Figure 2a and 2b, the mechanism is almost the same, that is, the light scattered by Au cluster or NP excites the Au film to generate a random SPP, which emits hot electrons via non-radiative plasmon decay. So why the measured photocurrent responses are so different for the two cases, with one gradually increases and the other almost maintains a certain value when it is "On"?

(2) The AuNPs generates hot electrons toward the TiO₂ layer and transfer to the Au film and water layer. With the illumination of light at on-resonance wavelength, will more and more positive charges concentrate at the AuNPs? The authors should give a more detailed discussion about this process.

(3) In line 139 Page 5, they explained the generation of positive burst-like photocurrent as the formation of a local electric field in the direction from the AuNPs toward the TiO₂. The local electric field prevents the diffusion of hot electrons in this direction, forming a local capacitor at the interface and causing the burst-like feature of the positive photocurrent. In contrast, the hot electrons generated by the inter-intraband transitions and random SPP damping in the infinite Au film are spilled out and participate in the reduction reaction at the TiO₂ – water interface.

However, in Figure 2a, the hot electrons were mainly generated by the SPP damping, was there also a burst-like photocurrent? Why it happens?

In Figure 4a, the hot electrons were also generated by the SPP damping, the lineshape of the photocurrent was changed as the sweep of wavelength. In some wavelengths, the burst-like phenomenon seems to vanish. Why?

They need to clarify the influence of the capacitor charging or discharging effects on the lineshape of the photocurrent, and also the influence of illumination wavelengths on the lineshape and capacitor charging or discharging effects.

(4) In Page 5 of Supplementary Information, they said "The offset of the photocurrent under the zero bias condition is attributed to the formation of the ultra-thin space-charge layer at the TiO₂-water interface." However, in Supplementary Figure 11, the measurement of I-V curve was under the non-illumination conditions, why it is called photocurrent?

(5) Some relevant references maybe discussed and compared with this work, such as Nano Lett., 2012, 12: 3808; Adv. Mater., 2014, 26, 6467-6471; and Laser & Photonics Reviews Doi:10.1002/Ipor.201600148

Reviewer #3 (Remarks to the Author):

Hot electron devices are currently a subject of active interest given their potentials applications ranging from photodetection to photocatalysis. This manuscript claims to study the interplay between hot electrons generated by two kinds of plasmons in a device: localized plasmons in gold nanoparticles and propagating plasmons on the surface of a gold film. Under different illumination conditions (namely different wavelengths of light), one or the other is predominantly excited, and leads to electrons (hence current) being injected in one direction or the other. Hence the device outlined in this paper shows wavelength-dependent polarity of the photocurrent (and photovoltage): blue-green wavelengths give a negative photocurrent while red illumination gives a positive photocurrent. This at least is how the authors interpret their results. I disagree with this picture as explained below.

The experimental results detailed in the manuscript are mostly solid and rigorous, to the extent that they show the polarity reversal of photoresponse unambiguously. The explanations for the mechanisms behind this behaviour however are less clear. The authors seem to muddle concepts regarding Rayleigh scattering and excitation of plasmons aided by this scattering. They also oversell their work in terms of potential technological applications as well as the new science it addresses. For example, line 53-55:

1. "the process of hot electron generation in thin-film coupled plasmonic nanostructures still remains to be understood. This process is of special interest because of the differences between the "localizing" and "propagating" properties of the LSPR and SPPs respectively."

This is an empty sentence. It conveys nothing to me. As far as plasmonic hot electron generation is concerned, I am not aware if different rules apply for localized versus propagating plasmons. Have the authors found otherwise? If so, what?

2. Likewise lines 91-92: "The Au film is used to promote the launching of SPPs at the interface between the TiO₂ layer and the Au film."

What does this mean? There is no surface plasmon without the Au film.

3. Lines 117-118. "The incident light is Rayleigh-scattered by the Au cluster. The scattered light excites the Au film to generate a random SPP at the interface with TiO₂, which subsequently emits hot electrons..."

I find it hard to believe that <3nm Au clusters positioned 25nm away from a Au film scatter enough light to launch SPPs on the Au film. Do the authors have any data supporting this claim other than photocurrent enhancement? Why can't the increase in negative photocurrent with <3nm clusters be explained by simply direct hot electron injection from the Au cluster  TiO₂  water? How can the authors be confident that increased SPP generation is indeed at play?

4. Lines 126-129: "As in the first two cases, a random SPP is generated and decays by emitting hot electrons. Simultaneously, the AuNP emits a large number of hot electrons... and omnidirectional light..., which also induces the generation of another random SPP."

This doesn't make sense. A random SPP in the Au film cannot be excited directly without Au particles (momentum mismatch). Only inter-intraband transitions in Au film can happen. Therefore, in the absence of scattering from AuNPs ("omnidirectional light") there are for sure no SPPs. Moreover, I am not convinced that scattering from 3-4nm Au particles can effectively launch SPPs at the Au-TiO₂ interface.

There is no evidence of SPPs even in the numerical simulations (Supplementary Figure 5).

To summarize, I believe that the authors see a wavelength-dependent polarity of photocurrent, and that this is due to an interplay between absorption in the Au film versus Au nanoparticles. I do not believe that I have seen evidence of surface plasmons being involved at all. Hence I find the basic premise of the paper not sufficiently supported by evidence.

To Reviewer 1:

We are grateful to the Reviewer for taking the time to review our manuscript and give us his/her valuable comments. We have taken all the comments into consideration and have made appropriate changes to the manuscript. Our point-by-point response appears below, in which we first echo the Reviewer's comments (shown in italic) and then respond to them.

Reviewer 1's comment #1:

Overall, this is a very nice work investigating the interplay between hot electrons generated from LSPRs and SPPs. The authors leveraged their understanding on this interplay to demonstrate a wavelength-controlled polarity-switchable photoconductivity device. To my best knowledge, this is the very first report of a device of this type. The experiments are well-designed and the data are overall well-presented and –interpreted. Given the interesting fundamental science unraveled in this work as well as some potential applications of the polarity-switchable device, I see a strong potential of this manuscript to be published as an article in Nat. Comm. Yet there are many detailed revisions that the authors might want to perform before the decision can be made.

Authors' response:

We thank the Reviewer for the positive evaluation of our work.

Reviewer 1's comment #2:

The authors mentioned many times (e.g. line97-98) that the Au film could be excited by the inter-intraband transitions. This statement is somewhat ambiguous. I remember that the interband onset of bulk Au is ~2.4 eV, below which the excitation is dominated by intraband transitions. The authors should present their data more rigorously according to this fact.

Authors' response:

We agree with the Reviewer's comment on the onset of the interband and intraband transitions of bulk Au. In this work, we characterized the devices in a large spectral region (400 nm - 900 nm). The interband transitions of Au dominate the electronic excitation at wavelengths below 500 nm (~2.4 eV) while the intraband transitions of Au dominate at wavelengths above 500 nm (~2.4 eV). We used the term "inter-intraband transitions" as a short notation for this relation.

To address the Reviewer's comment, we have made modifications to the manuscript (Page 4, Paragraph 1) such that in the revised manuscript, we have the following sentence: "Illumination with visible light excites electrons via interband transitions ($\lambda < 500$ nm) or intraband transitions ($500 \text{ nm} < \lambda < 700$ nm) in the Au film." We have, furthermore, replaced the term "inter-intraband transitions" with "interband or intraband transitions" in the remaining part of the revised manuscript.

Reviewer 1's comment #3:

Line 98, the authors claimed that "excited electrons can directly surpass the Au – TiO₂ barrier (1.7 eV) [21] and contribute to the net photocurrent via the reduction reaction at the TiO₂ – water interface", which to me is not rigorous. While some of the electrons can indeed travel ballistically through the TiO₂ layer, the authors have to acknowledge that the electrons can also be intercepted by the Au NPs embedded in the TiO₂. Due to the band structure, Au NPs are actually excellent electron sinks. This raises a related question. Have the authors attempted to quantify the quantum efficiency of the device? Due to the interception effect I mentioned, I suspect that the QE should be extremely low.

Authors' response:

We thank the Reviewer for this valuable comment. We agree with him/her that the electrons are partially intercepted by the AuNPs embedded in the TiO₂ layer which reduces the quantum efficiency. As he/she mentions, the quantum efficiency of the device is low, compared to commercially available devices. The incident-photon-to-electron conversion efficiency (IPCE) is approximately 0.1% at a wavelength of 550 nm (negative photocurrent) and 0.02% at a wavelength of 700 nm (positive photocurrent). The IPCE values of this device are within the order of that of the plasmonic devices reported in a previous report [R1].

To address the Reviewer's comment, we have added the following sentence to the revised manuscript (Page 4, Paragraph 1). "During this process, the electrons can be partially intercepted by the AuNPs embedded in the TiO₂ which reduces the quantum efficiency." Also, we note that the Schottky barrier height between TiO₂ and Au is ~

1 eV, as reported in literature [R2, R3], and have corrected the value accordingly.

Reviewer 1’s comment #4:

What is the purpose of Fig. 2a? Fig. 2b&c seem to make a nice comparison already. To me, Fig. 2a is redundant, especially because Fig. 4 will present the same information. Another question for Fig. 2, are the photocurrent figures real data or just schemes? I haven’t seen any clarifications on this. If they are real data, why does the photocurrent decay so fast when the light is on and switch the sign when the light is off in Fig. 2a, but not in Fig. 2b?

Authors’ response:

We thank the Reviewer for the comment. Regarding the first part of the comment, while Figure 4 depicts the structures of the fabricated devices, it does not contain information about the physical mechanism of hot-electron generation. Figure 2a, in comparison, illustrates the mechanism that is present in our devices. However, as the Reviewer points out, it looks redundant to the reader. To address this concern, we have deleted Figure 2a and its related content.

As for the second part of the comment, the photocurrents presented in Figure 2a, Figure 2b, and Figure 2c in the original manuscript are from our actual experimental results. The fast decay of the photocurrent in Figure 2a is due to a charging effect as the Au/TiO₂/electrolyte solution works as a capacitor based on the low conductivity of TiO₂. As a piece of proof of the charging effect, the discharge current was measured when the light irradiation was turned off. However, such a charging-discharging effect does not occur in the device with the active layers composed of AuNPs as shown in Figure 2b. We attribute this observation to the interception effect that the Reviewer mentions in his/her comment #3. It means that the device with the larger AuNPs shows a stronger interception effect than the device with the Au clusters or with neither the AuNPs nor the clusters. Due to this effect, the discharged electrons from the water – TiO₂ interface could not reach the Au film electrode, and hence, the discharge current was not present.

Response Figure 1 | Influence of the interception effect on the photocurrent of the device. **a**, Schematic of the dual-plasmon device presented in the manuscript. The AuNPs (or clusters) are positioned between the Au electrode and the Pt electrode. **b**, Schematic of a random SPP device with AuNPs (or clusters) that only scatter the incident light and do not contribute hot electrons to the readout photocurrent. **c** and **d**, Photocurrent measurements of the devices presented in **a** and **b**, respectively. The later device has 25 times larger active area than the former one.

To verify this explanation experimentally, we fabricated two different types of devices and measured their

photocurrent responses, as presented in Response Figure 1 above. Response Figure 1a shows a schematic of the dual-plasmon device as presented in the manuscript. Response Figure 1b shows another type of the device whose AuNPs or clusters act as scattering centers only. The hot electrons generated by them do not contribute to the photocurrents. Therefore, the interception effect is excluded in this design. We measured this device using the two-electrode setting shorting the counter and the reference electrodes of an electrochemical analyzer. The photocurrent measurements of the devices at a wavelength of 600 nm are presented in Response Figure 1c and Response Figure 1d, respectively. It is clearly seen that, while the capacitor effect appears in the same way for the devices with the Au clusters or the bare TiO₂ layer, the device with the AuNPs shows a different behavior. In detail, the device with 2 nm AuNPs presented in Response Figure 1c does not have the charging-discharging effect, but the device presented in Response Figure 1d clearly exhibits this feature. This is evidence for the influence of the interception effect that we obtained in our study.

Reviewer 1's comment #5:

Line 139-142, the authors state that "Here, once the hot electrons are spilled out by the internal photoemission owing to the damping of the LSPR, a local electric field is formed in the direction from the Au NPs toward the TiO₂ and prevents the diffusion of hot electrons in this direction, forming a local capacitor at this interface and causing the burst-like feature of the positive photocurrent". While this is reasonable explanation for the data in Fig. 3a, why don't we see the same photocurrent burst in Fig. 5. The photocurrent in Fig. 5 is solely generated by the hot electron transfer from LSPRs, which should also give a current burst.

Authors' response:

We thank the Reviewer for giving us the opportunity to address this point. We are aware that not only Figure 3a but also Figure 5b indicate the small charging/discharging feature of photocurrents. In our experiment, this current burst also happens in the device whose AuNPs are located on top of the TiO₂ layer and are in direct contact with the water electrolyte [R4]. The current bursts appear under illumination and lead to quasi-static photocurrents in such devices [R4].

Since we feel that our explanation for Figure 3a in the original manuscript is not sufficient, we explain the current burst as follows. In the case of Figure 3a, the positive photocurrent from the AuNPs and the negative photocurrent from the Au film come into play. While the positive photocurrent has the burst-like feature, as described above, the negative photocurrent is more stable under visible illumination. This interplay effect induces the strong burst-like feature of the photocurrent, as presented in Figure 3a.

To address the Reviewer's comment, we have made modifications to the manuscript (Page 5, Paragraph 2) such that in the revised manuscript, we have the following sentences: "In the case of Figure 3a, the positive photocurrent from the AuNPs and the negative photocurrent from the Au film come into play. While the positive photocurrent has the burst-like feature, the negative photocurrent is more stable under visible illumination. This interplay effect induces the strong burst-like feature of the photocurrent, as presented in Figure 3a. Meanwhile, the hot electrons generated by the interband or intraband transitions and random SPP damping in the infinite Au film are spilled out and participate in the reduction reaction at the TiO₂ – water interface."

Reviewer 1's comment #6:

What is the reason for the current resonance peak at ~600 nm in Fig. 4g? Here, the photocurrent either results from direct optical excitation of Au film (Fig. 4a) or scattered light excitation (Fig. 4b&c). I don't see any resonance mechanisms in these processes.

Authors' response:

We thank the Reviewer for raising this point. We agree with the Reviewer that there is no resonance in the AuNPs here, as presented in the optical spectra in Figure 4f. In our understanding, the optical absorption by the defect-induced electronic levels in the TiO₂ film can also generate a photocurrent which may partially compensate for the photocurrent from the Au film, forming the dip in the photocurrent response of the device presented in Figure 4g. In addition, we investigated the photocurrent of another random SPP device that is presented in Response Figure 1b. As shown in the following discussion (presented in Response Figure 4), the dip in the photocurrent is very broad and located between the wavelength of 450 nm and 550 nm.

To address this comment, we have added the following sentences to the revised manuscript (Page 7, Paragraph 1) "In Figure 4g, similar dips located at the wavelength of around 600 nm are observed in the photocurrent measurements for the three cases. These dips are not related to the plasmon resonance as the TiO₂ film with and without the clusters does not exhibit any resonance, as presented in Figure 4f. However, the defect-induced

electronic levels in the TiO₂ film can generate a photocurrent which partially compensates for the photocurrent from the Au film. Therefore, the dips are present in the photocurrent response."

Reviewer 1's comment #7:

Fig. 6a, why don't we see the polarity switching on the curve of 5 nm Au NPs?

Authors' response:

We thank the Reviewer for the comment. In Figure 6a, we intend to show the dependency of the photocurrent on the particle size. The parameter given in the figure is the thickness of the Au film that was deposited to the device, not the particle's diameter (as described in Figure 6a's caption). Specifically, the deposition of the 0.3 nm-thick Au film results in the formation of ~2.5 nm-diameter Au clusters, while the deposition of the 2 nm-thick Au film forms ~4 nm-diameter AuNPs. Additionally, the deposition of the 5 nm-thick Au film induces the formation of an infinite Au film. This infinite Au film does not show any resonant peak, as presented in Supplementary Figure 7 in the revised Supplementary Information. Therefore, the hot electrons induced by the LSPR are absent. In this case, the polarity switching does not occur.

Reviewer 1's comment #8:

The Method section is repeated in the SI. The authors should rearrange it.

Authors' response:

We thank the Reviewer for the comment. To address the reviewer's comment, we have removed the Method section from the Supplementary Information.

To Reviewer 2:

We are grateful to the Reviewer for taking the time to review our manuscript and give us his/her valuable comments. We have taken all the comments into consideration and have made appropriate changes to the manuscript. Our point-by-point response appears below, in which we first echo the Reviewer's comments (shown in italic) and then respond to them.

Reviewer 2's comment #1:

The authors realized the simultaneous generation of localized and propagating plasmon-induced hot electrons and investigated their cooperative interplay in a metal-semiconductor device. And they also demonstrated such a device as a wavelength-controlled polarity-switchable photoconductivity. According to theoretical analyses and experimental verifications, they elucidated that the polarity of the dual-plasmon device is determined by the balance in population and directionality between the hot electrons generated from LSPR and SPPs. Also, they demonstrated that the functionality and performance can be tailored by changing the design of the device or the applied bias potential.

It is a good and novel work, both experiments and theoretical analyses are sufficient. The presented results are interesting. Therefore, I will recommend this paper to be published in Nature Communications with possibly minor revisions based on the comment below:

Authors' response:

We thank the Reviewer for the positive comment.

Reviewer 2's comment #2:

In Figure 2a and 2b, the mechanism is almost the same, that is, the light scattered by Au cluster or NP excites the Au film to generate a random SPP, which emits hot electrons via non-radiative plasmon decay. So why the measured photocurrent responses are so different for the two cases, with one gradually increases and the other almost maintains a certain value when it is "On"?

Authors' response:

We thank the Reviewer for the comment. As the Reviewer points out, the mechanisms presented in Figure 2a and Figure 2b may look almost the same to the reader. Therefore, we have removed Figure 2a and its related content from the manuscript. To address the comment of the Reviewer, we provide the explanation as follows.

The photocurrents presented in Figure 2a, Figure 2b, and Figure 2c are from our actual experimental results. The fast decay of the photocurrent in Figure 2a is due to a charging effect as the Au/TiO₂/electrolyte solution works as a capacitor based on the low conductivity of TiO₂. As a piece of proof of the charging effect, the discharge current was measured when the light irradiation was turned off. However, such a charging-discharging effect does not occur in the device with the active layers composed of AuNPs as shown in Figure 2b. We attribute this observation to the interception effect that the Reviewer 1 mentions in his comment #3. It means that the device with the larger AuNPs shows a stronger interception effect than the device with the Au clusters or with neither the AuNPs nor the clusters. Due to this effect, the discharged electrons from the water – TiO₂ interface could not reach the Au film electrode, and hence, the discharge current was not present.

To verify this explanation experimentally, we fabricated two different types of devices and measured their photocurrent responses, as presented in Response Figure 1 above. Response Figure 1a shows a schematic of the dual-plasmon device as presented in the manuscript. Response Figure 1b shows another type of the device whose AuNPs or clusters act as scattering centers only. The hot electrons generated by them do not contribute to the photocurrents. Therefore, the interception effect is excluded in this design. We measured this device using the two-electrode setting shorting the counter and the reference electrodes of an electrochemical analyzer. The photocurrent measurements of the devices at a wavelength of 600 nm are presented in Response Figure 1c and Response Figure 1d, respectively. It is clearly seen that, while the capacitor effect appears in the same way for the devices with the Au clusters or the bare TiO₂ layer, the device with the AuNPs shows a different behavior. In detail, the device with 2 nm AuNPs presented in Response Figure 1c does not have the charging-discharging effect, but the device presented in Response Figure 1d clearly exhibits this feature. This is evidence for the influence of the interception effect that we obtained in our study.

Response Figure 1 | Influence of the interception effect on the photocurrent of the device. **a**, Schematic of the dual-plasmon device presented in the manuscript. The AuNPs (or clusters) are positioned between the Au electrode and the Pt electrode. **b**, Schematic of a random SPP device with AuNPs (or clusters) that only scatter the incident light and do not contribute hot electrons to the readout photocurrent. **c** and **d**, Photocurrent measurements of the devices presented in **a** and **b**, respectively. The later device has 25 times larger active area than the former one.

Reviewer 2's comment #3:

The AuNPs generates hot electrons toward the TiO₂ layer and transfer to the Au film and water layer. With the illumination of light at on-resonance wavelength, will more and more positive charges concentrate at the AuNPs? The authors should give a more detailed discussion about this process.

Authors' response:

We thank the Reviewer for raising this point. As the Reviewer states, at the on-resonance wavelength, the positive charges are concentrated at the AuNPs. In our understanding, as stated in the manuscript (Lines 139-142), the generation of hot-electrons by internal photoemission owing to the damping of the LSPR forms a local electric field in the direction from the AuNPs to the TiO₂ which prevents the diffusion of hot electrons in this direction. Finally, there is a state of equilibrium between the generated hot electrons and the diffused hot electrons at the AuNPs. Therefore, no more charges are produced at the AuNPs.

To address the Reviewer's comment, we have added the following sentence to the revised manuscript (Page 5, Paragraph 1): "Finally, a state of equilibrium is reached between the generated hot electrons and the diffused hot electrons at the AuNPs such that no more charges are produced at the AuNPs."

Reviewer 2's comment #4:

In line 139 Page 5, they explained the generation of positive burst-like photocurrent as the formation of a local electric field in the direction from the AuNPs toward the TiO₂. The local electric field prevents the diffusion of hot electrons in this direction, forming a local capacitor at the interface and causing the burst-like feature of the positive photocurrent. In contrast, the hot electrons generated by the inter-intraband transitions and random SPP damping in the infinite Au film are spilled out and participate in the reduction reaction at the TiO₂ – water interface. However, in Figure 2a, the hot electrons were mainly generated by the SPP damping, was there also a burst-like photocurrent? Why it happens?

In Figure 4a, the hot electrons were also generated by the SPP damping, the lineshape of the photocurrent was

changed as the sweep of wavelength. In some wavelengths, the burst-like phenomenon seems to vanish. Why?

They need to clarify the influence of the capacitor charging or discharging effects on the lineshape of the photocurrent, and also the influence of illumination wavelengths on the lineshape and capacitor charging or discharging effects.

Authors' response:

We thank the Reviewer for the comment. Regarding his/her first question about the burst-like photocurrent presented in Figure 2a, we attribute it to the interception effect. In response to comment #2 of the Reviewer, we provide an explanation of this effect. We would appreciate it if the Reviewer could refer to our response to comment #2.

As for the second and third questions, in our understanding, the charging-discharging effect occurs at the following interfaces: AuNPs – TiO₂, TiO₂ – water, and Au film – TiO₂. The discharging current is caused by the transport process of the electrons through all of the interfaces in the device. The quality of the TiO₂ layer, which reflects the density of the defect states, has a certain contribution to the capacitor effect.

To further study this effect in detail, we performed a control experiment using a device in which the TiO₂ layer was prepared by atomic layer deposition (ALD) instead of the sputtering method, as shown in Response Figure 2 below. Because the ALD method can deposit a high-quality TiO₂ film with higher crystallinity and fewer defects, the conductivity of the TiO₂ film deposited by using ALD is low. Here we compare the line shape of charging and discharging photocurrents with those in which the TiO₂ layer was prepared by the sputtering method. The measurement conditions were kept the same as were presented in the manuscript. Relatively shaper charging and discharging photocurrents were observed due to the lower conductivity of the TiO₂ layer as shown in Response Figure 2. Importantly, the line shape of the photocurrents stayed the same at all of the excited wavelengths. Therefore, the line shape of the photocurrents is not only wavelength-dependent, but also depends on the defect density and conductivity of the TiO₂ layer which is determined by the fabrication method.

Response Figure 2 | Dependency of the photocurrent's line shape on the quality of the TiO₂ layer. The TiO₂ active layer was prepared by the atomic layer deposition (ALD) method.

To address the Reviewer's questions, we have added the following sentence to the revised manuscript to describe

this effect (Page 7, Paragraph 1). "It is worthwhile to note that the lineshape of the photocurrents presented in Figure 4a shows a wavelength-dependent behavior. However, this feature also depends on the defect density of the TiO₂ layer. Specifically, for the device whose TiO₂ layer is prepared by atomic layer deposition which is known to have much smaller defect density than the sputtering method, such a feature disappears."

Reviewer 2's comment #5:

In Page 5 of Supplementary Information, they said "The offset of the photocurrent under the zero bias condition is attributed to the formation of the ultra-thin space-charge layer at the TiO₂-water interface." However, in Supplementary Figure 11, the measurement of I-V curve was under the non-illumination conditions, why it is called photocurrent?

Authors' response:

We thank the Reviewer for the comment. We agree with the Reviewer on this point. The I-V curve was under the non-illumination condition so that it has to be "current" instead of "photocurrent". We have corrected it in the revised manuscript.

Reviewer 2's comment #6:

Some relevant references maybe discussed and compared with this work, such as Nano Lett., 2012, 12: 3808; Adv. Mater., 2014, 26, 6467-6471; and Laser & Photonics Reviews Doi:10.1002/Ipor.201600148

Authors' response:

We thank the Reviewer for drawing our attention to these publications on the enhancement of the photoconversion efficiency. We think that in comparison with previous publications, while the primary similarity is the use of hot-electron generation from the designed plasmonic nanosystems, the primary difference is the generation of hot electrons from SPPs.

To address the Reviewer's comment and acknowledge the previous work, we have added the original work of a graphene-antenna sandwich photodetector (Zheyu Fang *et al.*, *Nano Lett.* **12**, 3808–3813 (2012)) and the hot-electron induced phase transition of a MoS₂ monolayer (Yimin Kang *et al.*, *Adv. Mater.*, **26** 6467–6471 (2014)) to the references in the revised manuscript.

To Reviewer 3:

We are grateful to the Reviewer for taking the time to review our manuscript and give us his/her valuable comments. We have taken all the comments into consideration and have made appropriate changes to the manuscript. Our point-by-point response appears below, in which we first echo the Reviewer's comments (shown in italic) and then respond to them.

Reviewer 3's comment #1:

Hot electron devices are currently a subject of active interest given their potentials applications ranging from photodetection to photocatalysis. This manuscript claims to study the interplay between hot electrons generated by two kinds of plasmons in a device: localized plasmons in gold nanoparticles and propagating plasmons on the surface of a gold film. Under different illumination conditions (namely different wavelengths of light), one or the other is predominantly excited, and leads to electrons (hence current) being injected in one direction or the other. Hence the device outlined in this paper shows wavelength-dependent polarity of the photocurrent (and photovoltage): blue-green wavelegths give a negative photocurrent while red illumination gives a positive photocurrent. This at least is how the authors interpret their results. I disagree with this picture as explained below.

The experimental results detailed in the manuscript are mostly solid and rigourous, to the extent that they show the polarity reversal of photoresponse unambiguously. The explanations for the mechanisms behind this behaviour however are less clear. The authors seem to muddle concepts regarding Rayleigh scattering and excitation of plasmons aided by this scattering. They also oversell their work in terms of potential technological applications as well as the new science it addresses. For example, line 53-55:

Authors' response:

We thank the Reviewer for the comment. We feel that the Reviewer's main concern is about the mechanism of exciting random SPPs in our device. In response to the concern, we have conducted additional experiments and computational analysis to support our explanation of the mechanism. We provide the results and explanation in our response to the Reviewer's comments #3 and #4 below.

Regarding the Reviewer's concern about overselling, it is not our intention to oversell our work. We think that the proposed applications stated in the manuscript are our perspective and expectation of the future potential use of the device and are not easily justified at this early stage of the study. On the other hand, we would like to suggest such potential applications to readers who may find our device useful for their future studies. However, in response to the Reviewer's comment to the sentences in line 53-55 in the manuscript, we have softened the statement in the revised manuscript such that it reads "Given the simplicity of the device, it may be useful as a replacement for current optical instruments such as wavemeters as well as other devices that hold complex functionalities, but suffer from a bulky and impractical setup. The presented dual-plasmon device may allow for the miniaturization and simplified control of such setups and enhance the applicability and accessibility of optoelectronic methods to research areas such as bio- and photo-chemistry."

Reviewer 3's comment #2:

1. "the process of hot electron generation in thin-film coupled plasmonic nanostructures still remains to be understood. This process is of special interest because of the differences between the "localizing" and "propagating" properties of the LSPR and SPPs respectively."

This is an empty sentence. It conveys nothing to me. As far as plasmonic hot electron generation is concerned, I am not aware if different rules apply for localized versus propagating plasmons. Have the authors found otherwise? If so, what?

Authors' response:

We thank the Reviewer for the comment. We do not think anyone knows the existence of different rules for localized versus propagating plasmons. What we mean by this above statement is that while the hot-electron generation mechanisms may be identical or different, the properties of the LSPR and SPPs are different because the former is localized and the latter is propagating. However, we think that the statement may sound misleading and confuse the reader as the Reviewer points out. Therefore, we have revised the statement in the manuscript such that it reads "Consequently, the process of hot-electron generation in film-coupled plasmonic nanostructures still remains to be understood. A better understanding of this process is expected to foster the development of new technologies for generating and controlling plasmon-induced hot-electrons and to facilitate the development of

novel plasmonic devices.”

Reviewer 3’s comment #3:

Likewise lines 91-92: "The Au film is used to promote the launching of SPPs at the interface between the TiO₂ layer and the Au film."

What does this mean? There is no surface plasmon without the Au film.

Authors’ response:

We thank the Reviewer for the comment. The sentence means that SPPs are launched at the interface between the TiO₂ layer and the Au film. As the Reviewer correctly states, there is no surface plasmon without the Au film. It is essentially identical to what we wanted to state. However, the statement seems ambiguous and confusing. Therefore, we have revised the sentence such that it now reads “The Au film is used to launch SPPs at the interface between the TiO₂ layer and the Au film.”

Reviewer 3’s comment #4:

Lines 117-118. "The incident light is Rayleigh-scattered by the Au cluster. The scattered light excites the Au film to generate a random SPP at the interface with TiO₂, which subsequently emits hot electrons..."

I find it hard to believe that <3nm Au clusters positioned 25nm away from a Au film scatter enough light to launch SPPs on the Au film. Do the authors have any data supporting this claim other than photocurrent enhancement? Why can't the increase in negative photocurrent with <3nm clusters be explained by simply direct hot electron injection from the Au cluster  TiO₂  water? How can the authors be confident that increased SPP generation is indeed at play?

Authors’ response:

We thank the Reviewer for the comment. To provide additional data to support our claim, we have conducted an additional experiment to verify the event that the Au clusters positioned away from the Au film can scatter enough light to launch SPPs on the Au film. In Response Figure 3 below, we show that our newly designed device for this purpose allows for Rayleigh scattering by the Au clusters or AuNPs to launch random SPPs. The existence of the SPPs is expected on both sides of the 20 nm-thick Au film as the Au film is semitransparent. Although the Au clusters and AuNPs are present in the device, their hot electrons do not contribute to the photocurrents. Experimental evidence can be found in form of the collected data displayed in Response Figure 4 below.

In Response Figure 4, we show the current responsivity of the device presented in Response Figure 3 over the entire spectral range. Similar to the data presented in Figure 4 in the manuscript, the photocurrent increases when Rayleigh scattering comes into play. Specifically, the resonance of the AuNPs (for the case of the 1 nm-thick Au film deposited on the TiO₂ layer) does not cause a significant increase to the net photocurrent. This device shows the current responsivity to be one order of magnitude smaller than that of the device presented in Figure 4g in the manuscript. This is attributed to the screening of the SPPs at the backside of the 20 nm-thick Au film. The positive photocurrent obtained at a wavelength of 400 nm is attributed to the defect-induced transitions in the TiO₂ layer which has twice the thickness of the dual-plasmon device.

Response Figure 3 | Another type of device for random SPP generation using Rayleigh scattering. a, Schematic of the device. **b,** Principle for the SPP excitation by Rayleigh scattering and generation of hot electrons via the decay of random SPPs. **c,** Picture of the device during the measurement.

Response Figure 4 | Current responsivity of the device shown in Response Figure 3.

As for the Reviewer's other concern, in our case, the Au clusters with sufficiently small diameters (approximately 2.5 nm) do not display LSPR, as presented in Figure 4. Therefore, the contribution of hot electrons by the Au clusters is excluded. Additionally, the influence of the catalytic activity of the Au clusters on the oxidation of the water is also eliminated as they are capped inside the TiO₂ layer. Therefore, the increase in the negative photocurrent with < 3 nm-diameter clusters cannot be explained by the direct hot-electron injection through Au cluster → TiO₂ → water.

Reviewer 3's comment #5:

Lines 126-129: "As in the first two cases, a random SPP is generated and decays by emitting hot electrons.

Simultaneously, the AuNP emits a large number of hot electrons... and omnidirectional light..., which also induces the generation of another random SPP."

This doesn't make sense. A random SPP in the Au film cannot be excited directly without Au particles (momentum mismatch). Only inter-intraband transitions in Au film can happen. Therefore, in the absence of scattering from AuNPs ("omnidirectional light") there are for sure no SPPs. Moreover, I am not convinced that scattering from 3-4nm Au particles can effectively launch SPPs at the Au- TiO₂ interface.

There is no evidence of SPPs even in the numerical simulations (Supplementary Figure 5).

To summarize, I believe that the authors see a wavelength-dependent polarity of photocurrent, and that this is due to an interplay between absorption in the Au film versus Au nanoparticles. I do not believe that I have seen evidence of surface plasmons being involved at all. Hence I find the basic premise of the paper not sufficiently supported by evidence.

Authors' response:

We thank the Reviewer for the comment. The major concern of the Reviewer is about the launching of the SPPs at the Au film - TiO₂ layer interface via Rayleigh scattering. To provide evidence for the generation of SPPs on the Au film, we have revised the presentation of the simulation results in the original manuscript and have conducted additional computational analysis as shown below.

In our simulations in the original manuscript (Supplementary Figure 5b and Supplementary Figure 5c), the absorption profiles under two different excitation wavelengths ($\lambda = 489$ nm which corresponds to the off-LSPR wavelength of the AuNP and $\lambda = 698$ nm which corresponds to on-LSPR wavelength of the AuNP) are presented. The color scale of these figures was originally set to emphasize the absorption intensity at the AuNPs at the expense of showing the absorption intensity in the Au film and hence made it difficult to see the evidence for the SPPs in the figure. To respond to the Reviewer's concern, an adjusted color scale allows us to see the absorption at the Au film - TiO₂ layer interface, as shown in Response Figure 5 below. Due to this adjustment, the color around the AuNP is oversaturated and not shown in the figure.

Response Figure 5 | Absorption profile of the dual-plasmon device. The device was simulated at an illumination wavelength of 698 nm (on-LSPR wavelength of the AuNP), as presented in Supplementary Figure 5. Here we show that the absorption intensity is evident at the Au film - TiO₂ layer interface after adjusting the color scale.

In Response Figure 5, the absorption at the Au film - TiO₂ layer interface is higher than that of the surrounding area, indicating field enhancement at this interface. The reason for this field enhancement is either from the existence of the SPPs at the interface or the penetration of the incident light to the Au film (here the penetration depth of the visible light in the Au film is ~ 20 nm [R5]).

In order to clarify this point, we have performed an additional FDTD simulation to investigate the power flow in the device. Response Figure 6 below presents the result of the simulation of a device consisting of a 2.5 nm-diameter cluster embedded in the TiO₂ layer. The distance between the cluster and the Au film is 20 nm. The light source is a pulse-like plane wave normal to the Au film with a wavelength of 635 nm. The grid size is non-uniform, being 0.488 nm around the Au cluster and 2 nm at other areas. The dielectric function of the Au and TiO₂ is taken from literature [R6, R7]. In Response Figure 6, the simulation of the Poynting vectors in the device is presented. In the figure, the direction of the Poynting vectors is from the Au film towards the Au cluster as the incident light is reflected by the Au film (after ~1.8 fs after the light hits the Au cluster). As the Poynting vectors reflect the power flow in the device, the non-uniformity of the Poynting vectors obtained in this simulation indicates the existence of the SPPs at the Au film - TiO₂ layer interface.

Response Figure 6 | FDTD simulation of the power flow in the device. It is presented as a snapshot at 1.8 fs after the incident light hits the device. The direction of the power flow in the device is indicated by the direction of the Poynting vectors. For clarity, the boundaries of the Au cluster - TiO₂ and Au film - TiO₂ layer are highlighted in blue.

To address the Reviewer's comment, we have modified Supplementary Figure 5 and added Response Figure 6 and related text to the revised Supplementary Information.

REFERENCES

- [R1] Knight, W. M., Wang, Y., Urban, A. S., Sobhani, A., Zheng, B. Y., Nordlander, P., & Halas, N. J. Embedding plasmonic nanostructure diodes enhances hot electron emission. *Nano Letters* **13**, 1687-1692 (2013).
- [R2] Mubeen, S., Hernandez-Sosa, G., Moses, D., Lee, J., & Moskovits, M. Plasmonic photosensitization of a wide band gap semiconductor: converting plasmons to charge carriers. *Nano Letters* **11**, 5548-5552 (2011).
- [R3] McFarland, E. W. & Tang, J. A photovoltaic device structure based on internal electron emission. *Nature* **421**, 616-618 (2003).
- [R4] Shi, X., Ueno, K., Oshikiri, T. & Misawa, H. Improvement of plasmon-enhanced photocurrent generation by interference of TiO₂ thin film. *The Journal of Physical Chemistry C* **117**, 24733-24739 (2013).
- [R5] Read, T., Olkhov, R. V., & Shaw, A. M. Measurement of the localised plasmon penetration depth for gold nanoparticles using a non-invasive bio-stacking method. *Physical Chemistry Chemical Physics* **15**, 6122 (2013).
- [R6] Johnson, P., & Christy, R. Optical constants of noble metals. *Physical Review B* **6**, 4730-4739 (1972).
- [R7] <http://www.spectra.com/sopra.html> (data of TiO₂ B). Date of access: August 23, 2016.

REVIEWERS' COMMENTS:

Reviewer #1 (Remarks to the Author):

The authors have now fully addressed my concerns and hence I recommend the publication of this manuscript on Nat. Comm.

Reviewer #2 (Remarks to the Author):

The author addressed all of my questions, and thus I suggest accepting the work for publication in Nature Communications.

Reviewer #3 (Remarks to the Author):

The authors have responded to the comments and criticisms of all the three reviewers and submitted their revised manuscript. I am satisfied with the explanations, additional measurements and the changes made to the manuscript. In particular, the new FDTD simulations show that even 2nm Au clusters can help launch plasmons at a nearby gold interface. In light of this, I recommend publication of the revised manuscript in Nature Communications.